# Primary and Acquired Resistance to Immunotherapy in Lung Cancer: Unveiling the Mechanisms Underlying of Immune Checkpoint Blockade Therapy

**DOI:** 10.3390/cancers12123729

**Published:** 2020-12-11

**Authors:** Laura Boyero, Amparo Sánchez-Gastaldo, Miriam Alonso, José Francisco Noguera-Uclés, Sonia Molina-Pinelo, Reyes Bernabé-Caro

**Affiliations:** 1Institute of Biomedicine of Seville (IBiS) (HUVR, CSIC, Universidad de Sevilla), 41013 Seville, Spain; lboyero-ibis@us.es (L.B.); josnogucl@alum.us.es (J.F.N.-U.); 2Medical Oncology Department, Hospital Universitario Virgen del Rocio, 41013 Seville, Spain; amparo.sanchez.gastaldo.sspa@juntadeandalucia.es (A.S.-G.); miriam.alonso.sspa@juntadeandalucia.es (M.A.); 3Centro de Investigación Biomédica en Red de Cáncer (CIBERONC), 28029 Madrid, Spain

**Keywords:** lung cancer, immunotherapy, resistance mechanisms, PD-1/PD-L1, immune checkpoint inhibitors, monoclonal antibodies, NSCLC, SCLC

## Abstract

**Simple Summary:**

Immuno-oncology has redefined the treatment of lung cancer, with the ultimate goal being the reactivation of the anti-tumor immune response. This has led to the development of several therapeutic strategies focused in this direction. However, a high percentage of lung cancer patients do not respond to these therapies or their responses are transient. Here, we summarized the impact of immunotherapy on lung cancer patients in the latest clinical trials conducted on this disease. As well as the mechanisms of primary and acquired resistance to immunotherapy in this disease.

**Abstract:**

After several decades without maintained responses or long-term survival of patients with lung cancer, novel therapies have emerged as a hopeful milestone in this research field. The appearance of immunotherapy, especially immune checkpoint inhibitors, has improved both the overall survival and quality of life of patients, many of whom are diagnosed late when classical treatments are ineffective. Despite these unprecedented results, a high percentage of patients do not respond initially to treatment or relapse after a period of response. This is due to resistance mechanisms, which require understanding in order to prevent them and develop strategies to overcome them and increase the number of patients who can benefit from immunotherapy. This review highlights the current knowledge of the mechanisms and their involvement in resistance to immunotherapy in lung cancer, such as aberrations in tumor neoantigen burden, effector T-cell infiltration in the tumor microenvironment (TME), epigenetic modulation, the transcriptional signature, signaling pathways, T-cell exhaustion, and the microbiome. Further research dissecting intratumor and host heterogeneity is necessary to provide answers regarding the immunotherapy response and develop more effective treatments for lung cancer.

## 1. Introduction

Lung cancer is the most common cancer, contributing 11.6% of the total case number, and is responsible for 18.4% of cancer-related deaths around the world [1,2]. Despite the social effort to identify high-risk populations by lung cancer screening and the development of prediction tools, as well as prevention campaigns, the incidence of lung cancer is expected to increase by 71.4% by 2040 worldwide [3]. It has been estimated that approximately 1 in 15 people will develop lung cancer throughout their lives [4]. In addition, according to sex, it is the most prevalent tumor in men and the third most prevalent tumor in women. Although lung cancer causes more deaths every year than all breast, prostate and colorectal tumors combined according to the American Cancer Society, the good news is that the decline in lung cancer mortality has been accelerating by 2% in the last decade for both men and women [4].

Tobacco smoking is the leading risk factor associated with this disease, with 80% of cases attributed to it in Western countries. Nevertheless, this pattern shows up to 20-fold variation in lung cancer rates from one country to another, especially according to the level of development and socioeconomic status of the region, where the rates of pollution also play an important role [5]. Thus, it is more common in developed countries, especially in the USA and Europe, and less frequent in less developed countries, such as Africa and South America [6]. Moreover, exposure to secondhand smoke, radon gas, asbestos, infections and genetic susceptibility are other common risk factors for lung cancer [2]. Individual susceptibility, which derives from different human polymorphisms present in the human population, affects the balance between metabolic activation, detoxification, and reparation of DNA adducts differently [7,8].

Lung cancer is a heterogeneous group of malignant tumors of epithelial cells originating in the lining or glandular epithelium of the bronchial tree, with a common cellular and molecular origin but with different accumulated genetic alterations and different clinical and prognostic behaviors. This heterogeneity among patients has led to the establishment of different subgroups according to morphological, immunohistochemical and genetic characteristics. Thus, lung cancer is classified into two main groups, each ranging from stage I to IV depending on tumor progression: small cell lung cancer (SCLC) (15% of patients) and non-small cell lung cancer (NSCLC) (80% of patients). The latter is, in turn, subdivided into three subtypes: large cell carcinoma (LCC, 10% of all cases), squamous cell carcinoma (SCC, 25%), and lung adenocarcinoma (ADC, 40%). Between the main histological subtypes of NSCLC, studies have shown that SCC tends to arise centrally within the main or lobar bronchus, showing slow growth, and its prevalence is highly associated with tobacco smoking, which increases the mutational burden by 10 times [9]. In these tumors, the proposed actionable genes with clinical efficacy are limited. On the other hand, ADC appears more peripheral, mainly affecting distal bronchioli and alveoli, with glandular and mucin differentiation. ADC is the most common subtype among never smokers and women [10], and although it generally has a worse prognosis, it is frequently associated with druggable driver mutations such as epidermal growth factor receptor (EGFR) mutation and echinoderm microtubule associated protein-like 4 (EML4)-anaplastic lymphoma kinase (ALK) fusion protein, among others, for which there are targeted therapies with good clinical results [2]. In turn, SCLC is frequently located centrally in the lung and presents a poor prognosis. Consequently, the implementation of genetic characteristics for the histological classification of cancer opens the possibility of developing novel targeted therapies and therefore optimizing precision medicine [11].

## 2. Clinical Management of Patients with Lung Cancer

The treatment of lung cancer depends on several factors, such as the condition of the patients, histological features, and tumor TNM staging. Classical treatment options include surgical resection, platinum-based chemotherapy and radiation therapy, in monotherapy or in combination, as well as sequential therapeutic strategies, among others. Surgery is the standard option for early-stage lung cancer, when the cancer is confined to the lung and therefore considered to be curable [12]. These stages involve potentially resectable lesions, macroscopically or microscopically, offering more guarantees of control or cure of the disease [13]. In intermediate stages, surgery is only for diagnostic purposes because at this point, the disease becomes difficult to control [14]. Despite surgery, the recurrence rate remains high in the early stages of NSCLC, and 30–55% of patients with curative resection develop recurrences that occur mainly at a distance [15]. In very advanced cases, that is, stages associated with distant metastases, surgery only has indication with palliative or diagnostic effects, and systemic anticancer treatments are applied in an attempt to slow tumor growth and improve overall survival (OS) and the quality of life of the patient and minimize symptoms [16]. Regarding systemic treatments, chemotherapy (CT) is the first option in patients with SCLC. However, in patients with NSCLC, it is used as a complementary strategy to surgery, where it can be given before (neoadjuvant) or after (adjuvant) surgery for curative or palliative purposes. In early-stage NSCLC, radiotherapy (RT) can be used instead of surgery with curative effects. After surgery in stage II and IIIA patients, adjuvant CT has proven to prevent recurrence [17], while unresectable stage III lung tumors are recommended to be treated with CT-RT. In the case of metastatic cancer, treatment with CT has been established, but RT is also used for palliative care of symptoms [18,19]. In the case of SCLC, treatment consists of a combination of platinum-based CT with etoposide. In addition, RT will be used in cases of localized disease and for the prevention and treatment of brain metastases.

However, despite the results obtained with CT and RT in lung cancer, they have many undesirable side effects, and the mortality and morbidity data are still very high, with 5-year OS rates of approximately 10-15%, making it the most lethal neoplasm [1]. Approximately 25% of lung cancer cases harbor genetic abnormalities amenable to treatment with targeted therapies, improving the survival time of patients. This has allowed targeted therapies to appear in clinical oncology practice, improving the survival rates and quality of life of patients. These therapies are treatments that selectively target cancer-specific genes, proteins, or the tissue environment that contribute to cancer growth and survival, blocking them and minimizing damage to healthy cells [20]. Many of these targeted therapies focus on various tyrosine kinase receptors (RTKs) that are involved in cell growth and survival, whose alterations amplify the signals that lead to tumorigenesis. Thus, RTK inhibitors (TKIs) are used to interrupt these signaling pathways, but their use tends to produce acquired resistance [21]. Successful biomarkers with therapeutic purposes in patients with ADC have been mainly EGFR mutations, ALK- and RET- (rearranged during transfection) gene rearrangements, ROS1 (receptor tyrosine kinase ROS proto-oncogene 1) fusions; V600E-specific mutation of BRAF (v-Raf murine sarcoma viral oncogene homolog B) gene; MET (hepatocyte growth factor receptor) factor amplification, KRAS and HER2 (human epidermal growth factor receptor 2) gene mutations, among others [16,21,22,23]. In contrast, patients with SCC show few alterations for therapeutic purposes. FGFR1 (fibroblast growth factor receptor 1) amplification or PI3K (phosphoinositol 3-kinase) mutations are challenging targets but with no effective inhibitors yet [24]. However, the limited understanding of SCLC molecular biology has led to completely nonexistent targeted treatment today. In addition, it is important to highlight that targeted therapies in patients with lung cancer may fail or have little or no efficacy when treatments are carried out in populations that are not molecularly selected.

The greatest challenge for lung cancer clinical management has become markedly evident with the use of immunotherapy (IT). There is increasing evidence of the role of the host immune system in immunosurveillance and tumor rejection in which every known innate and adaptive immune effector mechanism participates. Moreover, may be transitory or long-lasting a crucial role so that local immunosuppression in a tumor context caused by chronic inflammation has protumor effects, whereas enhancing T-cell function and dendritic cell maturation derived from acute inflammation has the opposite (antitumor) effect [25]. Thus, the immune system has been recognized as another important cancer hallmark [26]. This has led to the development of therapeutic strategies focused in this direction, such as molecules for reactivating the host immune response as cancer vaccines or checkpoint inhibitors to emphasize intrinsic antitumor immunity [27]. However, a high percentage of patients with lung cancer do not respond to these treatments, or their responses may be transitory or long-lasting. Here, we review the relevance of immunotherapy in lung cancer, with a focus on the underlying resistance mechanisms of the disease.

## 3. Immunotherapy for Lung Cancer

Immuno-oncology has emerged as a promising field that seeks to reinforce the host’s own immune system to avoid immune evasion of the tumor by recognition of tumor-specific antigens (neoantigens) and tumor-associated antigens (TAAs) that trigger an immune reaction that causes tumor remission [28]. Therefore, attacking tumor cells directly is no longer the target of the therapy, but the strategy is redirected towards the immune system. This makes it work for any tumor histology or driver mutation, and the side effects are different from other therapies [29]. Accordingly, cancer immunotherapy has been developed based on several approaches, ranging from stimulating effector mechanisms to counteracting inhibitory and suppressive mechanisms. Vaccines are among the strategies to activate immune effector cells, while additional stimulation strategies include cytokines, adoptive cell therapy and oncolytic viruses. On the other hand, the use of antibodies against immune checkpoint molecules stands out as one of the strategies to neutralize immunosuppressive mechanisms [30] (Figure 1).

Thus, since the activation of the immune checkpoint pathways is a very frequent tumor evasion mechanism, the use of inhibitors against immune checkpoints (ICIs) arises as IT, stimulating antitumor responses towards tumor-specific antigens within the TME, which is composed of stromal cells, immune cells, and extracellular matrix, all of which are closely related to tumor cells. The rate of tumor-infiltrating immune cells in the TME can be classified as immunodesert (noninfiltrated or also called ‘immunologically cold’), immunoexcluded (peripheral immune infiltration around tumor cells) and immunoinflammed (infiltrated or also called ‘immunologically hot’). The latter expresses immune checkpoint molecules and is correlated with a more favorable response to ICI therapy [31].

Because IT has promising results in terms of survival and quality of life, more treatments of this type for lung cancer are trying to be developed, since it is frequently diagnosed in advanced stages, when other classical therapies such as surgery, chemotherapy, and radiotherapy are minimally effective. Nonetheless, IT can be combined with any of the therapeutic modalities already described. While the synergistic effects of combining these drugs are being evaluated, they do not produce additional toxicities or death [29].

### 3.1. Nonspecific Immunotherapies

Nonspecific immunotherapies, also known as immunomodulatory therapies, are not directed against a specific antigen but rather aim to bolster the antitumor immune response. They usually involve both the innate and adaptive responses of the immune system, and their mechanisms of action can be direct antitumor effects, reversing immunosuppression, activating innate immunity, and activating antigen-nonspecific T-cells [32]. These treatments include cytokines (interleukins and interferons) as the most frequently used compounds, although there are others, such as immune-stimulatory agents (CpG oligonucleotides or *Bacille Calmette-Guérin*, BCG), antibodies and enzyme inhibitors [32]. The two Food and Drug Administration (FDA)-approved cytokines for the treatment of severe malignancies are IL-2 and IFN-α; however, neither of them has an indication in lung cancer.

### 3.2. Oncolytic Viruses (OVs)

Tumors create an immunosuppressed microenvironment that allows them to escape from the immune system, but in turn, this makes tumors more sensitive to viral infections. Oncolytic virotherapy, which uses attenuated and genetically modified viruses capable of selectively infecting tumor cells, is based on this premise, taking advantage of the deregulated pathways to produce cell lysis. The second mechanism of action of OVs is the induction of antitumor immunity, which takes place thanks to the antigens released during oncolysis. The third is the ability to produce acute vascular-disrupting effects that lead to tumor reduction [33]. Thus, in OV therapy, the virus is the active agent itself and not a carrier, as occurs in gene therapy [34]. The advantages of oncoviral immunotherapy are its specificity targeting tumor cells, its independence of specific receptor expression patterns and therefore of the associated resistances, and its ability to enhance the antitumor immune response or induce a novel nonself antigen response [35].

Some of the viruses that are being considered for this type of immunotherapy are from the herpes simplex family, such as fowlpox virus, Newcastle disease virus, reovirus and measles virus (MV), but some adenoviruses, picornaviruses (including coxsackie), reovirus, maraba, vaccinia virus, retroviruses and mumps are also considered [35,36,37]. In 2015, the FDA approved talimogene laherparepvec (T-VEC or Imlygic), a second-generation oncolytic herpes simplex virus type 1 (HSV-1) armed with Granulocyte Macrophage colony-stimulating factor (GM-CSF) [34], for the treatment of metastatic melanoma, which was the first approved oncolytic viral immunotherapy, although there are several clinical trials of oncolytic viruses that cover almost all solid tumors, including lung cancer.

Today, there are some OVs garnering intense interest for use in lung cancer clinical practice. This is the case for TG4010, which is a modified vaccinia virus Ankara designed to express MUC-1 and IL-2 that is being evaluated in a phase III clinical trial for advanced NSCLC. In addition, it has been reported that TG4010 in combination with CT improves the progression-free survival of patients and increases durable responses and long-term survival. Furthermore, there is evidence pointing to a synergistic effect when combined with anti-PD-1/PD-L1 (programmed cell death protein 1 and programmed death ligand 1, respectively) ICIs [37,38,39]. On the other hand, the oncolytic vaccine MAGE-A3 is also being evaluated in two phase I/II clinical trials. One of them uses the MG1 Maraba/MAGE-A3 (MG1MA3) virus alone or in combination with the adenovirus/MAGE-A3 (AdMA3) virus in patients with incurable advanced MAGE-A3-expressing solid tumors (NCT02285816; https://clinicaltrials.gov). Another clinical trial evaluated the combination of MG1MA3 + AdMA3 + pembrolizumab in previously treated patients with metastatic NSCLC (NCT02879760; https://clinicaltrials.gov). Inoculation of the antigen-expressing adenovirus that is also encoded within the OV results in T-cell-mediated tumor clearing and protection from relapses *in vivo*, avoiding an antiviral response [40]. Additionally, coxsackievirus A21 (CVA21, CAVATAK), which targets ICAM-1 naturally and in combination with pembrolizumab, is well tolerated and appears to increase the number of PD-L1+ tumor cells in a phase Ib clinical trial [41]. Another coxsackievirus, type B3 (CV-B3), is a nonenveloped, human-pathogenic enterovirus that produces a significant reduction in cell survival in KRAS-mutant NSCLC cell lines [42]. Adenovirus AVID-317 has shown effectiveness in 70% of tested human NSCLC-derived cell lines and an increase in the median survival in murine models [43]. Similarly, infection of NSCLC cells by MV induces tumor apoptosis in vitro and reduces tumor size in mice [44]. In SCLC, there are also OV studies, specifically with a modified oncolytic myxoma virus (MYXV), in murine models. This strategy produces tumor-specific cytotoxicity, necrosis mediated by immune cell infiltration and increased survival [45]. Despite these promising results, further studies are needed to optimize the delivery of vectors, as well as to obtain OVs with precise coordination with the immune system to effectively eradicate the tumor.

### 3.3. Adoptive T-Cell Immunotherapy

In adoptive cell transfer, tumor-reactive lymphocytes from the patient are collected, cultured ex vivo and reinfused, often along with growth factors, into the patient as therapy with the goal of recognizing, targeting, and destroying tumor cells. The cytotoxic T lymphocytes (CTLs) that can be used for this are tumor-infiltrating lymphocytes (TILs), T-cell receptor (TCR)-modified T-cells and chimeric antigen receptor (CAR)-modified T-cells [46,47,48]. In the latter, the cells are genetically engineered to target tumor-specific surface antigens. The advantages of this strategy [48] are that it is highly specific towards tumor cells; has a robust clonal expansion ability; presents tropism towards the antigen, so it can migrate towards metastasis; generates memory, maintaining long-term effectiveness; and under the right therapeutic conditions, adoptive cell therapy can eradicate tumors.

The FDA approved the use of CAR-T-cells, making them the first genetically engineered modified cell therapy with this approval. However, despite the good results in hematological tumors, their application in solid tumors has been largely limited. This is believed to be due to difficulties in finding specific targetable antigens, T-cell homing, infiltration and survival in the TME [47]. Nevertheless, strategies are already being developed to overcome these obstacles, such as the split, universal, and programmable (SUPRA) CAR system, which uses universal receptors that allow target multiplexing and implements multiple advanced logic and control features [49]. The application of this technology has not been very extensive in the treatment of patients with NSCLC, caused among other reasons by the fact that choosing an appropriate target is a challenge. Notwithstanding, there are currently 5 ongoing clinical trials (NCT04153799, NCT03525782, NCT03029273, NCT04025216 and NCT02706392; https://clinicaltrials.gov) trying to validate these new treatment options. However, despite being a powerful tool in nonresponsive patients and immunologically cold tumors, it is a high-cost individualized medication, which is currently limiting its use in the general population.

### 3.4. Cancer Vaccines

The idea of using cancer vaccines emerged in the late 19th century from the observation that some tumors remitted spontaneously after the patient suffered an infectious disease. Thus, vaccines are designed as an active specific immunotherapy that stimulates the immune response by presenting a pathogen or TAAs and producing an adaptive antitumor response. This boosts tumor antibodies and T-cells in vivo in a similar way to passive immunotherapy (i.e., tumor-specific antibodies or T-cells) [50]. In addition, they can be used for both treatment and prevention and may or may not be combined with other therapies.

Several types of tumor vaccines have been investigated, such as (*i*) allogeneic vaccines in which antigens come from non-self-cancer cells to stimulate the cytotoxic immune response; (*ii*) antigens or protein-based vaccines; (*iii*) autologous dendritic cell vaccines in which self-dendritic cells (DCs) are activated with tumor antigens; iv) DNA vaccines in which an expression plasmid harbors the target antigen; and (*v*) vector-based vaccines in which antigens are administered through special viruses, bacteria, yeast cells or other structures. The most widely used vaccines are peptide-based vaccines with immunogenic epitopes, usually from tumor-specific or tumor-associated antigens. This strategy typically uses synthetic peptides, DNA or RNA to encode the neoantigen; however, the fact that these neoantigens are not universal for a type of tumor or a group of patients limits their widespread use and leads to the development of tailored and even polyneoantigen vaccines [47].

Despite efforts, only one vaccine has been approved by the FDA, namely, the therapeutic DC-based cancer vaccine Sipuleucel-T (Provenge™), for the treatment of metastatic castration-resistant prostate cancer [51]. However, in some Latin American countries, the CIMAvax- Epidermal Growth Factor (EGF) and Racotumomab vaccines have been approved for advanced NSCLC [46]. Despite the fact that the use of CIMAvax-EGF in NSCLC seems to have a tendency towards clinical benefit and to be immunogenic, the phase III randomized trials carried out have not been able to demonstrate sufficient efficacy and survival impact to include it in the protocols of NSCLC treatment [52]. In the case of racotumomab, approximately 20–25% of patients who received this therapeutic vaccine in a phase II/III study did appear to have great clinical benefit with longer progression-free survival (PFS) and OS [53,54]. However, these strategies are not supported as clinical approaches to patients with lung cancer.

### 3.5. Monoclonal Antibodies (mAbs)

Monoclonal antibodies recognize a single epitope region by a pair of variable domains (fragment of antigen binding, Fab) [55]. Antibodies are involved in cell lysis, which is caused by antibody-dependent cell-mediated cytotoxicity (ADCC) carried out by immune effector cells such as natural killer cells, neutrophils, mononuclear phagocytes and DCs. In addition, antibodies are also key in complement-dependent cytotoxicity (CDC) and the induction of adaptive immune responses, which intervene in the long-term benefit of these treatments through the presentation of tumor-derived peptides on MHC class II molecules (activating CD4^+^ T-cells) and on MHC class I molecules (activating CD8^+^ cytotoxic T-cells).

mAbs are produced as chimeric, humanized or human antibodies by recombinant DNA hybridomas. This is done to avoid immunogenicity of the original murine antibodies, which decrease the efficacy due to the human anti-mouse antibody response [51]. Depending on the type of antibody, a different suffix will be added to the name of the treatment. Thus, murine mAbs will use the suffix -omab, chimeric mAbs will end in -ximab, humanized mAbs in -zumab and human mAbs in -umab [56]. In turn, mAbs can be self-acting nacked (the most frequent in nonleukemic cancers), having a therapeutic function by targeting growth factor receptors, or conjugated, when they are combined with chemotherapeutic drugs or radioactive isotopes. Nacked mAbs work either by labeling tumor cells, by targeting immune system checkpoints or by blocking tumor antigens involved in cell growth and spread. Meanwhile, the conjugated mAbs act as a specific delivery system. On the other hand, there are bispecific mAbs capable of recognizing two different epitopes, normally one in tumor cells and the other in immune effector cells, which bring them closer together. To overcome its clinical limitations due to their short half-life and toxicity, a modification known as BiTE (bispecific T-cell engager) molecules has been developed, as well as engineered protein scaffolds with antitumor activity [55].

Different mechanisms of action have been attributed to mAbs with antitumor effects (Table 1 and Table 2). First, they can function as targeted therapy when designed against a specific tumor molecular target. In this case, when the type and location of the tumor do not define the treatment but depend only on the molecular target, it is called tumor-agnostic treatment. The first drug approved by the FDA with this indication was pembrolizumab, an anti-PD-1 antibody used to treat patients with unresectable metastatic solid tumors with microsatellite instability-high (MSI-H) or DNA mismatch repair deficiency (dMMR). Second, mAbs can activate the patient’s immune system to destroy tumor cells. This usually consists of unblocking certain pathways that are altered as a tumor evasion mechanism by targeting immunoregulatory coreceptors, reversing tumor immunosuppression and modulating the constant fragment (Fc) domain of mAbs. Ipilimumab, a cytotoxic T lymphocyte-associated antigen 4 (CTLA-4)-specific mAb that enhances the effector functions of T-cells while inhibiting Regulatory T cells (Tregs), stands out among these immune cell-targeting therapeutic strategies in NSCLC. However, there are also other approaches for different tumors, including antibodies targeting CD40, CD25, CD134, CD137, etc. (Table 1 and Table 2). Fc domain modifications are accomplished by mutations resulting in improved ADCC, such as in ocrelizumab (anti-CD20), or by modification of the oligosaccharide content (defucosylation) [55]. Finally, blocking ligand binding to growth factor receptors and/or their signaling pathways, as well as targeting the tumor microenvironment, through the inhibition of angiogenesis, cytokines, and growth factors, are other therapeutic strategies with increasing chances of success [55]. Examples of these latter types of antibodies are nintedanib, a triple angiokinase inhibitor used as a second-line therapy in combination with taxotere, and bevacizumab (a humanized VEGF-specific mAb), which was the first approved agent against tumor angiogenesis [57].

Currently, neutralizing monoclonal antibodies targeting immune checkpoints such as CTLA-4 and PD-1/PD-L1 have shown significant efficacy against various types of cancer, including NSCLC. Anti-CTLA-4 was the first immune checkpoint antagonist available for NSCLC; however, it has shown higher toxicity and less effectiveness than anti-PD-1/PD-L1 treatments, the latter being the most successful to date. Currently, four of these ICIs have been approved for NSCLC: nivolumab and pembrolizumab (both anti-PD-1) and atezolizumab and durvalumab (both anti-PD-L1) [28] (Table 1).

In the case of SCLC, ICIs appear to be a promising therapy, and numerous clinical trials are underway to test them (Table 2). Thanks to the results of the CheckMate-032 study, the FDA approved the use of nivolumab as a third-line therapy for metastatic SCLC in 2018. The following year, atezolizumab with carboplatin and etoposide as first-line therapy and durvalumab in combination with CT in 2020 were approved for extensive-stage SCLC by the FDA [58].

Despite the great advances achieved through the use of ICIs, not all patients with lung cancer respond to this treatment due to primary resistance and the development of secondary resistance. As a result, long-lasting clinical remission only represents a small percentage of outcomes. For this reason, great effort is being made to find predictive and monitoring biomarkers [28].

## 4. Impact of Immunotherapy on the Survival of Patients with Lung Cancer

Despite substantial improvements in survival rates achieved by therapeutic advances in solid tumors since 1975, in the case of lung cancer, platinum-based chemotherapy has been the only therapeutic option with limited benefit on OS and very few long-term survivors [59]. Thus, the 5-year OS is only ~5% for patients with metastatic NSCLC, and between 20–25% or 2% for SCLC, depending on the extent of the disease. Although targeted therapies (i.e., against EGFR and EML4/ALK mutations) have subsequently improved these results and show response rates of 80%, only 20–25% of worldwide patients are candidates for these therapies, and some of them will relapse and require additional therapeutic options.

Regarding modulation of the immune response as a therapeutic option, lung cancer has also found insufficient results with the use of nonspecific immunotherapies such as interleukins and interferons or, more recently, vaccines. However, monoclonal antibody-based immunotherapy has emerged, revolutionizing the treatment of these tumors. In particular, in patients with NSCLC, anti-PD-1/PD-L1 therapy has shown the greatest significant benefit in OS, long-term responses, and a good safety profile, including naïve and pretreated patients, regardless of the histological subtype. Patients with advanced NSCLC treated with nivolumab (anti-PD-1) in a phase I clinical trial showed a 5-year OS rate of 16%, quadrupling survival with standard CT [60]. These therapies were initially recommended for that subgroup of patients with high expression of PD-1 in tumors, who achieved better results in terms of survival with less toxicity [61]. However, it has been observed that ICI therapy even works in patients with tumors that are negative for PD-1 expression.

On the other hand, the results of the CheckMate-017/057 trials [62], in which patients with lung cancer were treated with nivolumab versus docetaxel as a second-line treatment, showed long-term clinical benefit for both SCC (23% versus 8% 2-year OS, respectively) and non-SCC NSCLC (29% versus 16% 2-year OS, respectively) [63]. Similarly, the KEYNOTE-010 study showed that in patients with advanced NSCLC treated with pembrolizumab, the OS was greater than that with docetaxel (12.7 vs 8.5 months, hazard ratio (HR) 0.61, 95% confidence interval (CI) 0.49–0.75; *p* < 0.0001). This difference was even more pronounced in patients with at least 50% of tumor cells expressing PD-L1 (17.3 vs 8.2 months; HR = 0.50, 95% CI 0.36–0.70; *p* < 0.0001) [64]. Finally, the phase III OAK study showed that atezolizumab in previously treated patients with NSCLC improved OS compared to docetaxel (13.8 vs 9.6 months; HR 0.73, 95% CI 0.62–0.87, *p* = 0.0003), and furthermore, this benefit was independent of histology and PD-L1 [65,66].

Currently, the most commonly used indication is first-line monotherapy with pembrolizumab for patients with NSCLC with >50% PD-L1 expression (KEYNOTE-024) [67] and the combination of CT + pembrolizumab for those patients whose PD-L1 expression is <50% (KEYNOTE-189) [68]. There is also a current first-line indication for the use of CT combined with atezolizumab + antiangiogenic drugs (bevacizumab) due to the significant improvements in PFS and OS of this combination versus the standard-of-care bevacizumab + CT observed in the IMpower150 study [69].

Therapeutic combinations based on immunotherapies are also being evaluated, and their results are highly promising. For instance, a synergistic effect has been seen in the combination of anti-PD-1 and anti-CTLA-4, which may result in even more long-term responders and could continue to improve OS [29], although toxicity also increases in some cases. Specifically, this study showed an OS of 17.1 months with nivolumab + ipilimumab (95% CI 15.0–20.1) compared to CT, which was 14.9 months (95% CI 12.7–16.7) for patients with NSCLC with a PD-L1 expression level of 1% or more (*p* = 0.007). This benefit was also observed when PD-L1 expression was less than 1% (17.2 vs. 12.2 months) [70]. In addition, the effect continues in 70% of the patients who interrupt the treatment [71]. Recently, the 3-year update from CheckMate-227 showed that these new dual IT regimens without CT achieve OS above 30% regardless of PD-L1 expression, as well as a 3-year sustained response in one-third of responder patients [72]. Overall, it appears that ICIs are well tolerated in terms of quality of life compared to other cancer therapies [73].

With regard to SCLC, nivolumab was also tested for pretreated patients with SCLC in the CheckMate-032 trial, obtaining approval from the FDA. This trial showed an objective response rate (ORR) of 10% with nivolumab and 23% with nivolumab + ipilimumab, with grade 3–4 adverse effects of 14% and 33%, respectively [74]. Soon after, the addition of atezolizumab (anti-PD-L1) to CT in the first-line SCLC treatment in the IMpower133 trial achieved the first OS improvement in decades [70]. This analysis showed that the risk of death decreased by 30% with the combination versus CT alone, without deterioration of the safety profile (median OS: 12.3 vs. 10.3 months; HR = 0.70, 95% CI 0.54 to 0.91, *p* = 0.0069). A similar reduction in the death risk was also identified in the phase II CASPIAN trial, which included the PD-L1 inhibitor durvalumab in combination with CT [71]. Thanks to the results of the KEYNOTE-028 and KEYNOTE-158 trials, the FDA approved in 2019 the use of pembrolizumab as monotherapy for patients with metastatic SCLC with disease progression on or after platinum-based CT and at least one other prior line of therapy [74].

Besides, there are several trials currently ongoing evaluating different uses of ICIs is SCLC. Among them, the phase III CheckMate-331 trial (nivolumab vs topotecan/amrubicin), the phase III CheckMate-451 trial (nivolumab vs nivolumab + ipilimumab vs placebo), the phase III KEYNOTE-604 trial (pembrolizumab + platinum/etoposide vs platinum/etoposide), the phase III CASPIAN trial (durvalumab ± tremelimumab + CT vs CT) and the phase III MERU trial (Rova-T + dexamethasone vs placebo (after CT)) [74].

In addition, much of the current research on NSCLC is focusing on studying the combination of different ITs with each other, as well as CT, RT, and targeted gene and cell-based therapies, with encouraging results [61]. Specifically, encouraging results from the CheckMate-9LA (dual IT plus CT) trial were presented, although the follow-up time was still not very long (12 months) [75]. Several preclinical and clinical studies are also testing the combination of ICI with RT. This combination has been found to have synergistic effects in NSCLC, improving the survival of lung cancer patients without significantly increasing adverse reactions. For example, in the phase I KEYNOTE-001 trial, patients with advanced NSCLC who had received RT prior to pembrolizumab significantly improved their PFS (4.4 vs. 2.1 months) and OS (10.7 vs. 5.3 months), while toxicity remained similar to patients treated with pembrolizumab alone. However, the underlying mechanisms of this combined therapies, possible biomarkers, and optimal therapy parameters - especially the design of RT- have not yet been clarified [76,77].

Finally, there are other strategies that hold great promise against cancer, such as CAR T-cell and suicide gene therapies; in 2013, CAR T-cell therapy achieved a response rate of 89% in acute lymphoblastic leukemia and complete responses in acute B lymphoblastic leukemia, so its effect on solid tumors is currently being investigated. In the case of suicide gene therapy, it is based on the use of genes encoding toxic proteins or enzymes capable of transforming a prodrug into a toxin. Typically, adenovirus and herpes virus, among others, are carriers. The use these therapy systems would lead to the sudden and massive presentation of TAA that can be synergistically enhanced by its combination with ICI. The combination of both therapies has demonstrated its antitumor effect in murine models. However, this hypothesis is still in the preclinical stages of development [78].

## 5. Resistance to Immune Checkpoints in Lung Cancer Immunotherapy

Immunotherapy, specifically treatment with ICIs, has been implemented in the clinical routine as a standard of care, with special relevance in patients with NSCLC who have shown unprecedented durable response rates. However, it has been observed that the majority of patients do not respond initially to treatment or relapse after a period of response. This is due to resistance mechanisms, whose understanding is key to preventing them and increasing the number of patients who can benefit from these treatments. Furthermore, the scarcity of competent immune preclinical models in which tumor regression is induced by ICIs limits the study and understanding of the mechanisms involved in the response [79].

Resistances to ICIs can be classified into primary (or innate) and secondary (or acquired) resistance (Figure 2), and all them can be mediated by both intrinsic and extrinsic host factors. The former prevents the infiltration or function of immune cells in the TME, while the latter involves components other than tumor cells within the TME [80]. In primary resistance, patients do not respond to initial treatment with ICIs, mainly due to a lack of recognition by T-cells because of the absence of tumor antigens. In secondary resistance, patients relapse after a period of initial response as a consequence of the appearance of tumor evasion mechanisms. Primary resistance to IT accounts for 7–27% of first-line treatment and 20–44% of second-line treatment in patients with lung cancer [81]. According to the KEYNOTE-001 trial results, approximately 25% of patients treated with ICIs could develop secondary resistance [82]. It is a dynamic process in which the response depends on the immune/tumor cell balance [83]. The TME provides a chronic inflammatory and immunosuppressive space for tumor development [84], and it can manifest clinically as primary resistance, mixed responses or secondary resistance.

Intrinsic resistance mechanisms include genetic and epigenetic alterations that alter the formation, presentation and/or processing of neoantigens, as well as disruption of cellular signaling pathways that lead to impaired action of cytotoxic T-cells [85]. These mechanisms can be summarized in the absence of antigens or their aberrant processing, lack of antigen presentation (loss of human leukocyte antigen (HLA)), genetic T-cell exclusion and insensibility to T-cells. Some of the pathways altered in these mechanisms are the MAPK, PTEN, PI3K, WNT/β-catenin, STING and IFN-γ signaling pathways, as well as constitutive PD-L1 expression by cancer cells. On the other hand, extrinsic mechanisms involve noncancerous stromal or immune cells or other systemic influences, such as microbiota, that promote the tumor process [85]. The most relevant include the lack of or exhaustion of T-cells, immune checkpoint blockers (i.e., CTLA-4 or PD-1), and immunosuppressive cells. Here, the role of Tregs, myeloid-derived suppressor cells (MDSCs) and type II macrophages is highlighted, as well as that of regulatory molecules released in the TME, such as IFN-γ, IDO, CEACAM1, TIM-3, TGF-β or adenosine [80,86]. Other host factors that also influence treatment resistance include endocrine, metabolic, environmental (dysbiosis, antibiotic or steroid consumption) and personal factors (age, chronic disease or genetic susceptibility) [87]. Additionally, different resistance mechanisms (primary/secondary) and antitumor immunity converge depending on the tumor phenotype. Thus, immunodesert tumors show immunological ignorance, tolerance or lack of T-cell priming. Immunoexcluded tumors are able to evade stromal factors as a result of mechanical barriers, vascular factors or an immune-suppressive chemokine state. In the case of immunoinflammed tumors, all the mechanisms mentioned above come together [87]. In addition, Syn et al. [88] observed a parallelism between the primary and secondary resistance mechanisms.

Among the most remarkable resistance mechanisms in lung cancer are aberrations in tumor neoantigen burden, effector T-cell infiltration in the TME, epigenetic modulation, transcriptional signature, signaling pathways, T-cell exhaustion, and the microbiome. Their involvement in resistance to immunotherapy in this disease is described below (Figure 2).

### 5.1. Tumor Neoantigen Burden

The effectiveness of PD-1/PD-L1 blockade is also correlated with the tumor mutation burden (TMB) [89,90,91]. Tumors with high mutational loads, in the range of 5–10 somatic mutations per megabase of DNA, such as NSCLC [89] and SCLC [92,93], are highly immunogenic and correlate with a high ORR, extended PFS and/or durable benefit after PD-1 blocker treatment. In the case of SCLC, these tumors present fewer mutations per megabase than NSCLC, which could be related to the lower efficacy of immunomodulatory therapies observed in this histological subtype of lung cancer [94]. Thus, high immunogenicity increases the number of neoantigens and therefore the sensitivity to therapy through the development and infiltration of antigen-specific effector T-cells. This increase in the number of mutations is due to deficiencies in DNA damage repair pathways, which occur more frequently in tumors of patients with durable responses to anti-PD-1 therapy (21% of patients with complete responses and 53% with objective radiographic responses [95]) due to MSI-H [31]. C-to-A transversions and specific gene deletions are also correlated with clinical benefits [96]. In the case of lung cancer, tobacco smoke contains many carcinogens that produce a high-transversion mutational profile known as the “molecular smoking signature” [89] responsible for the increase in TMB and, therefore, for the efficacy of ICI treatment, especially in patients who smoke compared to never smokers [94].

Many studies have attempted to determine a TMB cutoff capable of identifying responders to ICI therapy. For example, Rizvi et al. [89] determined a cutoff of 178 mutations per tumor for NSCLC, Carbone et al. [97] set it at ≥ 243 mutations for patients with NSCLC from the phase 3 CheckMate-026 trial, while Hellmann et al. [91] established it at ≥ 248 mutations for patients with SCLC from the CheckMate-032 trial. Often, PD-L1 expression was not predictive of response and did not correlate with TMB [94], the latter being an independent positive predictive biomarker [98].

Selective loss of mutation-associated antigens by a T-cell-dependent immunoselection process has also been proposed as an immunoediting mechanism of cancer in acquired resistance. Furthermore, these neoantigens can be lost by gene downregulation, removal of tumor subclones, loss of mutated alleles or elimination of chromosomal regions harboring truncal alterations [84]. Thus, tumors with a poor response correlate with those whose neoantigens are expressed in a smaller number of tumor cells (subclonal level) [31]. In NSCLC, it has been observed that a T-cell response that has a favorable impact on clinical outcome requires clonal neoantigens present in 100% of the tumor cells, as it is associated with an inflamed TME. Meanwhile, subclonal neoantigens do not elicit an effective antitumor response [99].

On the other hand, tumor cell clones that never express neoantigens can proliferate, constituting another escape route [85]. Furthermore, the efficiency of the tumor in the presentation of neoantigens on MHC-I to T-cells also influences the response to ICIs, since the antitumor activity of CTLs depends on it. The efficiency of antigen presentation can be diminished by the reduction/loss of MHC-I expression, decreasing the affinity of neoantigens for MHC-I or the homozygosis of the HLA-A, HLA-B and HLA-C genes [96]. In early-stage untreated NSCLCs, multiple independent mechanisms of immune evasion have been observed due to strong selective pressure from the immune system. Specifically, it has been observed that clonal neoantigens may be subject to loss of copy number (through HLA-LOH (loss of heterozygosity)) or transcription downregulation (by hypermethylation of the promoter, which occurs in ~23% of neoantigens, or other additional mechanisms) [100]. Furthermore, the SCC subtype has been observed to express HLA class I genes and B2M to a lesser extent than adenocarcinoma and normal tissue. This would explain the lack of correlation between OS and neoantigen load observed in SCC tumors [99].

### 5.2. Effector T-Cell Infiltration in the TME

The response to PD-1/PD-L1 blockade is highly dependent on several mechanisms that alter the infiltration of effector T-cells in the TME, such as aberrations in neoantigen or T-cell activation or even inaccessibility of T-cells to the tumor. First, the number and diversity of previously activated tumor-specific T-cells, as well as the infiltration of effector T-cells in the TME, which is associated with mutational and neoantigen burden, especially if the antigens are derived from essential mutated proteins for tumors such as p53, are key. Accordingly, a correlation was observed between the density of TILs and survival in lung cancer. Thus, high levels of CD8+, CD3+ and CD4+ T lymphocyte infiltration in the TME showed better OS in patients with lung cancer [101], and T helper type 1 (Th1) polarization and consequent CD8+ T-cell activation also correlated with a stronger antitumor immune response [102,103]. Thus, a resistance mechanism consists of avoiding the infiltration of effector T-cells into the tumor using epigenetic mechanisms, modifications of the posttranscriptional program, alteration of chemokines and stromal cells and TIL dysregulation [96]. In relation to T-cell activation during PD-1 blockade, T-cell-DC crosstalk is required. Accordingly, a new immunotherapy strategy for NSCLC with DCs cocultured with cytokine-induced killer cells (DC-CIKs) has been extensively studied and demonstrated its efficacy and safety, showing a significant improvement in PFS, OS and disease control rates (DCRs) for patients with NSCLC without an increase in serious adverse events [104]. On the other hand, tumor-infiltrating DCs produce IL-12 in response to the IFN-γ produced by neighboring T-cells, which are in turn activated by IL-12. This forms a loop that can be amplified by the activation of the noncanonical NF-κB transcription factor pathway [105].

Second, stromal cells form a physical myofibroblastic barrier around tumor cells induced by TGF-β that prevents infiltration by influencing the migration and positioning of T-cells. In addition, cancer-associated fibroblasts, which are the main stromal cells and are associated with poor patient prognosis, can mediate T-cell death and dysfunction through PD-L2 and FAS antigen ligand (FASL) [31]. This has been shown in lung cancer, where PD-L2 and FASL are enriched within the stromal regions of tumors [106]. TGF-β can also be secreted, as well as other inhibitory cytokines such as IL-10 or IL-35, by Treg cells to prevent the immune response within the lung tumor environment [87] by inducing CD4^+^ T-cell differentiation into inducible Tregs (iTregs) [102].

### 5.3. Epigenetic Modulation

Epigenetics comprises inheritable and reversible changes in the genome without modifying nucleic acid sequences. Epigenetic mechanisms include DNA methylation, histone modifications, nucleosome remodeling, and alterations in noncoding RNA expression. Epigenetic alterations produce abnormal tumor-associated gene expression that causes tumorigenesis and cancer progression. Some of these epigenetic alterations have been associated with the efficacy of immunotherapy in vivo and in vitro [107] due to phenotypic changes in not only cancer cells but also immune cells for cellular killing and functional adjustment [108,109]. For example, epigenetic loss of the IFN-γ signaling pathway is related to resistance to anti-CTLA-4 treatment [109]. There are currently promising clinical trials combining epigenetic agents and immunotherapy as cancer therapy.

Alterations in epigenetic regulation affect the expression of immune checkpoints and TAAs in tumors, damaging antigen presentation, as well as the migration of T-cells to the TME, the cytokine profile and T-cell activation, inducing cell death [109]. These alterations consist mainly of methylation changes. The addition of methyl groups to DNA is generally a repressive label that prevents the expression of tumor suppressor and apoptosis genes. Epigenetic alterations are also frequently achieved by the acetylation of histones, which causes an increase in gene expression by relaxing chromatin, which in turn favors the recruitment of transcription factors [110]. Other epigenetic mechanisms associated with clinically beneficial responses to ICIs include disruption of the SWI/SNF complex, a chromatin remodeler, enhancing secretion of effector T-cell-attracting chemokines through increased tumor sensitivity to IFN-γ [96].

Thus, it has been shown that the promoter DNA methylation of PD-1, PD-L1, and CTLA-4 inversely correlates with the expression of these proteins in NSCLCs compared to normal tissues [111,112], as well as in other types of tumors. This is possibly mediated by TGF-β1 and TNF-α during EMT [113]. Furthermore, there seems to be an association between this type of epigenetic modification and the response to immunotherapy and survival in NSCLC [114]. This association can be determined by a signature of DNA methylation in 301 CpGs, namely, EPIMMUNE, which can be reduced to FOXP1 epigenetic status for clinical practice. FOXP1 is a transcription factor involved in quiescent CD4+ T-cell and follicular T helper cell regulation.

Treatment with epigenetic agents potentiates innate and adaptive immune pathways by improving antigen presentation, reexpressing TAAs, upregulating MHC class I and MHC class II, increasing IFN-*γ* release by tumor-specific CTLs, enhancing the proinflammatory functions of DCs to boost T-cell proliferation and effector T-cell trafficking, and improving the transcriptional activity of some genes related to the immune system, including PD-L1 and genes of the interferon signaling cascade [109].

These agents may also increase the number of TILs, which are decreased due to the suppression of Th1-type chemokine CXCL9 and CXCL10 expression, thereby enhancing the clinical efficacy of PD-L1 checkpoint blockade. They can also lead to the expression of a major T-cell costimulatory molecule on tumor cells, which in turn improves the immune response by efficient antitumor CTLs [115].

On the other hand, T-cell exhaustion arises as a consequence of persisting antigenic stimulation during chronic infections and cancer, producing effector impairment functions. However, anti-PD-1/PD-L1 treatment can reverse this situation, which is known as reinvigoration. Nevertheless, an acquired resistance mechanism to checkpoint inhibitors, which leads to a lack of long-lasting improvements, consists of the genetic stability of exhausted CD8+ T-cells, which prevents them from acquiring the memory T-cell phenotype after reinvigoration. The inability to acquire a memory T-cell phenotype despite transcriptional rewiring and reengagement of the effector circuitry is believed to be due to the epigenetic landscape, which is minimally remodeled after PD-L1 blockade [84].

It should not be forgotten that miRNAs also participate in all these processes. For example, miR-214, miR-126, and miR-568 participate in T-cell exhaustion by promoting the development of Tregs and improving their function, thus downregulating CTL activity [109]. In the case of NSCLC, the expression of PD-L1 is regulated by ZEB-1, which blocks the expression of miR-200, an inhibitor of PD-L1 expression [113].

### 5.4. The Innate Anti-PD-1 Resistance (IPRES) Gene Signature

A set of transcriptional signatures with the ability to stratify tumors, including lung adenocarcinoma, according to response to anti-PD-1 treatment (referred to as innate anti-PD-1 resistance, IPRES) has been reported [116]. This molecular classification displays concurrent deregulation of genes involved in increased hypoxia, angiogenesis, mesenchymal transition, and wound healing. Specifically, it consists of 26 transcriptomic signatures related to immunosuppression (IL10, VEGFA, and VEGFC), EMT transcription factors (AXL, ROR2, WNT5A, LOXL2, TWIST2, TAGLN, and FAP), monocyte and macrophage chemotaxis (CCL2, CCL7, CCL8 and CCL13), wound healing, angiogenesis and treatment/resistance to MAPK pathway inhibition [84,85,116]. It is suggested that mitigating the biological processes underlying this IPRES signature may reverse and modulate the efficiency of the ICIs.

### 5.5. PD-L1 Expression

PD-1-mediated T-cell suppression is a recurrent resistance mechanism. PD-L1 overexpression often exists in the TME, as well as in immune, stromal and tumor cells, and it is regulated by the latest response to oncogenic signaling or induced by inflammatory cytokines. In acquired immune resistance, TIL-secreted IFN-γ and tumor antigen-specific T-cells mediate PD-L1 upregulation after tumor antigen recognition, and this functions as a tumor escape mechanism that uses constitutive oncogenic signaling, such as c-Jun and STAT3.

On the other hand, the mutational landscape of the tumor can also influence PD-L1 expression, which may also be overexpressed as a consequence of amplification of the chromosomal region 9p24.1 (containing PD-L1, PD-L2, and JAK2) [85]. Then, PD-L1 binds to PD-1 on T-cells adjacent to tumor cells and sends a coinhibitory signal dampening T-cell activity [117]. Other mechanisms of PD-L1 upregulation are overexpression of Myc along with mutated KRAS; activating mutations of EGFR, KRAS or JAK2; and CMTM4 and CMTM6 posttranslational regulation in tumor cells and DCs, in addition to other posttranscriptional modifications [96]. According to the EGFR mutations and ALK rearrangements, the absence of high levels of CD8+ TILs and concurrent PD-L1 expression indicates innate resistance and limits the response to anti-PD-1/PD-L1 treatment in the majority of EGFR-mutant and ALK-positive NSCLCs. This PD-L1 expression can be due to constitutive oncogenic signaling [118]. Based on these results, PD-L1 expression has been proposed as an important predictor of the response to ICIs. In fact, the KEYNOTE-024 results demonstrate that treatment with pembrolizumab (an anti-PD-1 inhibitor) produces significantly longer PFS and OS and fewer adverse events when there is PD-L1 expression on at least 50% of tumor cells in advanced NSCLC [67]. This prompted the FDA to approve pembrolizumab as a first-line treatment for PD-L1+ (≥50%) NSCLC with no EGFR or ALK genomic tumor aberrations in 2016.

However, PD-L1 expression is not a good predictor of response, since it also depends on other factors, such as the presence of CD8+ TILs and the clonal TCR repertoire. Building on this, a tumor classification system [119] has been proposed where type I tumors are PD-L1-/TIL- (immune ignorance), type II tumors are PD-L1+/TIL+ (acquired immune resistance), type III tumors are PD-L1-/TIL+ (immune tolerance by other suppressor factors), and type IV tumors are PD-L1+/TIL- (constitutive expression of PD-L1) [120]. This tumor classification system, which is called TIME (Tumor Immunity in the MicroEnvironment), has been applied to colorectal cancer to predict response to immunotherapy without success [121]. However, studies in NSCLC highlight the role of TILs such as CD8+ FOXP3+ T-cells (a Treg subtype), CD8+ T-cells, and FOXP3+ T-cells. Thus, longer OS was associated with low tumor CD8+ FOXP3+ T-cell density in the TME, making it a likely negative prognostic factor in NSCLC. On the other hand, significantly more CD8+ T-cells were found in lung adenocarcinoma than in SCC. It should be noted that while adenocarcinomas showed better OS, SCCs expressed significantly higher amounts of PD-L1 [122].

### 5.6. T-Cell Exhaustion

T-cells play a key role in antitumor function and have demonstrated to be crucial for cancer immunotherapy. However, they are not completely effective because part of them go into a dysfunctional state of exhaustion. This promotes uncontrolled growth of tumors [123]. Under physiological conditions, T-cells specifically recognize and react to tumor antigens through their TCRs. Following T-cell priming and tumor localization, the balance between co-stimulation and co-inhibition determines degree of T-cell activation and subsequent immune response [124]. On the one hand, co-stimulation amplifies T-cells activation and enhances CTL proliferation, survival and effector function. On the other, inhibitory receptors, suppressive soluble mediators, cell subsets and metabolic factors from immunosuppressive TME lead to T cell “exhaustion” [123]. The intensity of these different signals depends on parameters such as specific mutations in cancer cells, spatial gradients in tumor composition, and therapy-induced alterations in TME. Collectively, these immunosuppressive signals in TME lead to the intratumoral T-cell exhaustion by influencing the expression of inhibitory receptors, changing metabolic pathways, modifying the epigenetic state, and altering their transcription factor profiles [125].

Functional exhaustion of CD8+ T-cells has been well described in chronic viral infections and in cancer [85]. Exhausted T-cells show poor effector function, express inhibitory receptors, and have altered transcriptional states. They are characterized by a hierarchical loss of proliferation and cytolytic activity, followed by defects in cytokine production and eventual depletion. In a first stage, interleukin-2 (IL-2) production and ex vivo killing capacity are lost, followed by loss of tumor necrosis factor-α (TNF-α) production, and ends with loss of interferon-γ (IFN-γ) and granzyme B (GzmB) production [126]. In addition, exhausted T-cells are associated with the overexpression of multiple inhibitory receptors, including PD-1, TIM-3, CTLA-4, LAG3 and BTLA, which in turn are associated with resistance to anti-PD-1 therapy in NSCLC. Depending on the expression of these five receptors in CD8+ T-cells, a depletion gradient is created that includes severe defects in cytokine production, proliferation and migration when all of them are coexpressed [84]. However, there are different subpopulations of PD-1+ CD8+ T-cells that respond differently to anti-PD-1 treatment in solid tumors. In addition, different epigenetic T-cell states influence the reprogrammability and dysfunction of exhausted PD-1+ T-cells [127]. A recent study conducted with NSCLC data and validated with pan-cancer data identified a 78-gene signature for exhausted CD8+ T-cells (GET) [128]. Furthermore, T-cell exhaustion was related to immune cytolytic activity. This could be explained by the understanding that T-cell exhaustion is a heavily regulated hyporesponsive adaptation to maintain prolonged tumor control while limiting detrimental immunopathology rather than just loss of functionality. However, the GET signature was not predictive of the clinical response to ICIs until it was combined with immune cytolytic activity (CYT) and T-cell-inflamed gene expression profiles (GEPs), which can be interpreted as T-cell exhaustion not implying complete immune incompetence [128].

Therefore, the scientific community has focused on the development of therapies that enhance T-cell anti-tumor immunity, including adoptive transfer of TILs, endogenous peripheral blood-derived T-cells (ETC), CAR-T, and TCR -engineered T-cells (TCR-T), neoantigen vaccines and checkpoint blockade therapies. To improve the efficacy of these therapies, the dysfunctional state of the T-cells must be reversed. This has led to the investigation of transcriptional regulators as well as metabolic and epigenetic factors as possible targets [123].

### 5.7. Genomic Drivers

Specific oncogenic signaling pathways play a crucial role in PD-1/PD-L1 inhibition resistance. An example is the increased activity of the PI3K-AKT pathway, which is commonly observed in many tumors. One of its causes is the loss of PTEN, which suppresses PI3K signaling activity, constituting a primary resistance mechanism, and constitutive expression of PD-L1. The lack of PTEN in tumor cells, which can be biallelic, decreases T-cell-mediated tumor killing, CD8+ T-cell infiltration into tumors, successful T-cell expansion from resected tumors and outcomes with PD-1 inhibitor therapy. This occurs through the upregulation of the expression of the immunosuppressive cytokines VEGF and CCL2 [84,85,96,129]. In the case of NSCLC, loss of PTEN expression occurs frequently (55% to 74%) [130]. Furthermore, the PTEN mutation rate in this pathology ranges from 8% to 17% [130], while promoter methylation occurs in up to 35% of NSCLC [131]. On the other hand, loss of heterozygosity at microsatellites surrounding and intragenic to the PTEN locus occurs in 19% [132]. It has also been proposed that activation of the WNT/β-catenin signaling pathway in many tumors is involved in primary resistance to anti-PD-L1/anti-CTLA-4 monoclonal antibody therapy [133]. Its activation is correlated with the absence of CD8+ T-cells and reduced CCL4 gene expression, leading to diminished infiltration of CD103+ dendritic cells and impaired antitumor immune responses [84,85,134]. On the other hand, dysfunctional mutations in JAK genes (JAK1 and JAK2) have been described as a secondary resistance mechanism. This loss of function was found in homozygosity by deletion of the wild-type allele. Although CD8+ T-cells produce IFN-γ after tumor recognition, complete loss of JAK functionality results in the absence of STAT1 phosphorylation and insensitivity to IFN-γ. Consequently, there was a drop in the expression of MHCI and PD-L1. Similarly, the knockdown of IFN-γ receptors (Ifrngr1 and Ifngr2) leads to resistance to anti-PD-1 treatment. However, the role of JAK1/2 could be more complex because its inhibition could also overcome resistance. Another mechanism of secondary resistance is truncating mutations in the β-2-microglobulin (B2M) gene. This loss results in impaired cell surface expression of MHC class I since B2M is key in its folding, stabilization and expression at the cell surface. Therefore, the presentation of antigens to cytotoxic T-cells is impaired [84,85,135]. B2M inactivation is enriched in patients who are nonresponsive to immune checkpoint blockade. Under other conditions, loss of STK11/LKB1 in the setting of an oncogenic KRAS mutation produces silenced STING expression and an inability to detect cytoplasmic double-stranded DNA (dsDNA). Its loss also increases IL-6, resulting in an increase in neutrophil recruitment, a decrease in T-cell infiltration, higher levels of T-cell exhaustion markers (PD-1, CTLA-4 and TIM3), and lower expression of PD-L1 on tumor cells [85,136], which impairs the ICI response. STK11/LKB1 mutations often coexist with KEAP1 mutations in NSCLC, correlating with a worse clinical outcome. Mutations in KEAP1, which have a frequency of 15–18% in NSCLC, produce constitutive activation of NRF2, leading to cellular resistance to oxidative stress, proliferation, and metabolic reprogramming [137]. Finally, T-cell recruitment and function can be impaired by MAPK pathway activation through activating point mutations of the BRAF gene [138]. BRAF mutations are detected in 3–8% of NSCLC, of which canonical V600E mutations represent approximately 50–75%, and they are found mainly among smokers. ICI treatment in patients with BRAF-mutated NSCLC seems to be more effective; however, its use in this context is not yet fully justified [139].

### 5.8. Enteric Microbiome

The microbiome is also suggested as a potentially key mechanism of immune resistance in patients with lung cancer. There is already scientific evidence showing the importance of the intestinal microbiota in the response to CT and IT and how its alteration and the concomitant use of antibiotics inhibit the benefit of ICIs in advanced cancer, decreasing OS and PFS in NSCLC [140]. Among the different immune cells, the microbiota has been shown to be associated with the development of effector cells of the immune system, such as Th1, Th2, Th17 and Treg cells [141,142,143]. In the case of SCLC, there is not yet evidence that supports the role of the microbiome specifically as an immune resistance mechanism.

With respect to microbiota composition, the relative abundance of *Akkermansia muciniphila* appears to significantly affect the clinical response to anti-PD-1/PD-L1 therapy in NSCLC and renal cell carcinoma. This was demonstrated with in vivo models by administering oral feces from ICI-responsive patients with NSCLC or oral supplements with *A. muciniphila* to restore the efficacy of PD-1 blockade. Furthermore, the authors propose that this effect is IL-12 dependent due to the increased recruitment of CCR9+ CXCR3+ CD4+ T lymphocytes [140]. In another study, a combination of 11 gut bacterial strains isolated from healthy donors was also capable of enhancing the antitumor efficacy of ICIs in mice by inducing IFN-γ-producing CD8+ T-cells in the intestine [96,144]. There are other similar studies where a greater diversity of the gut microbiome is related to the responding phenotype in melanoma models. Additionally, there is also a need for the presence of certain live bacteria, such as *Bifidobacterium*, to modulate DC activation and influence the T-cell response [145]. Taken together, these data suggest that resistance mechanisms to PD-1/PD-L1 blockade therapy are even more complex and require more understanding in lung cancer.

## 6. Potential Biomarkers of Resistance to Immunotherapy in Lung Cancer

One of the greatest challenges in the field of immuno-oncology is the identification of response/resistance predictive biomarkers, since only 20% of patients with NSCLC show a durable response to this type of treatment. The identification of new and reliable biomarkers able to discriminate between “responding” and “nonresponding” patients before starting therapy would allow us to treat only those patients who are going to have a clinical benefit with anti-PD1/PD-L1 therapy. This would mean not only an increase in the 5-year OS of patients with advanced NSCLC but would also allow us to not add toxicity to patients who are not going to benefit from the treatment and to provide economic savings due to the high cost of these treatments. In addition, longitudinal tumor sampling throughout immune checkpoint blockade treatment has emerged as a good methodological strategy to study the response and resistance mechanisms due to their dynamic nature. Thus, immune markers that did not show response prediction power before treatment did show response prediction power in the early stages of treatment [80,146].

To date, tumor PD-L1 expression and TMB have emerged as predictive biomarkers for NSCLC; however, both are imperfect tools, and this has led to further studies to find more effective tools. PD-L1 is an inducible and dynamic protein subject to changes according to the variations that exist in the TME. However, the predictive value of PD-L1 expression in immune cells and the TME for PD-1/PD-L1 blockade therapy has not yet been confirmed in NSCLC [147]. Similar results have been described for SCLC, where the extent of the benefits of PD-L1 expression on the long-term survival of patients continues to be demonstrated [148,149,150,151]. Its main limitations as a biomarker in immunotherapy are the variability between immunohistological techniques making standardization difficult, the tumor constitutive expression of PD-L1 driven by different mechanisms independent of the adaptive response to tumor immunity and TIL assessment. Thus, in addition to PD-L1 expression, strong peritumoral CD8+ T-cell activity is required to contribute significantly to the antitumor response. Together with the results of clinical trials, these findings suggest considering PD-L1 as a prognostic and not a predictive marker [152,153]. In the case of TMB, NSCLC and SCLC have a TMB average well above the rest of the tumors, with the exception of melanoma [93]. However, although their prognostic value is related to the number of neoantigens and the degree of immune infiltration, TMB cannot predict the immune infiltration in cancers driven by copy number alterations [31]. Moreover, TMB analysis through NGS platforms requires a large amount of tissue, and in some cancer types, substantial responses to ICIs are observed despite a low TMB. There are also studies [154] in which the median TMB did not differ between responders and nonresponders. However, the HLA-corrected TMB algorithm with HLA-LOH has been proposed as a solution since it shows additional predictive and prognostic value for response to ICIs [86] in advanced NSCLC. Recently, it has also been reported that the predictive power of TMB to differentiate the efficacy of ICIs is related to the age of the patients, i.e., it is better in young than elderly NSCLC patients [155]. Nevertheless, for TMB to be incorporated into clinical decisions, it is necessary to establish for which patient groups it would be beneficial, as well as the value of the cutoff for their benefit. For instance, Ricciuti et al. proposed that a classification of TMB by tertiles allows discerning patients with SCLC with better one-year PFS and OS rates [156]. However, these results need to be validated in large patient cohorts to extrapolate to all pre-ICI lung cancer patients.

Over the past few years, other potential TME predictive markers of response to ICI treatment in NSCLC have also been identified, such as the enzyme IDO1 (indoleamine 2,3-dioxygenases), which is overexpressed in NSCLC and catabolizes tryptophan into immunosuppressive metabolites such as kynurenine. IDO1 function is evaluated through the kynurenine/tryptophan ratio, which inversely correlates with PFS and OS. Patients with a lower ratio showed a better clinical outcome [152,157]. Therefore, IDO inhibitors in combination with ICIs are currently being conducted in several clinical trials including solid tumors, such as NSCLC and SCLC (NCT02298153, NCT03348904, NCT02959437, NCT03322566, NCT03322540, NCT03361228, NCT03347123, NCT03277352, and NCT03085914; https://clinicaltrials.gov). Additionally, the genetic mutations and gene expression signatures such as IPRES, described above, as well as the epithelial-to mesenchyme status in the TME, may have potential clinical use and aid in treatment decisions with ICIs, although further investigation of these biomarkers is required in NSCLC [147]. Otherwise, through NGS, the presence of MSI within a tumor, a signature of dMMR, and modifications in genes involved in response can also be evaluated, such as PTEN, STK11/LKB1, KEAP1, B2M, β-catenin and others described above [137]. In relation to this, dMMR and MSI-H have a high predictive value [96,115], while somatic mutations such as STK11/LKB1 and KEAP1 are prognostic of poor outcomes but not predictive [158] in terms of response to therapies with checkpoint inhibitors in lung adenocarcinoma.

There is a high percentage of patients with lung cancer who do not have adequate tissue at diagnosis for standard clinical tests, so it has been proposed to search for predictive biomarkers of response in peripheral blood. Blood is a type of sample that is frequently accessible, quickly and noninvasively, which allows (1) determining biomarkers in pretreatment; (2) continuous monitoring of the disease and treatment response evolution; and (3) avoiding comorbidities in patients by rebiopsy. Some of the potential blood markers are serum lactate dehydrogenase levels, the number of circulating tumor cells (CTCs), circulating tumor DNA (ctDNA), soluble forms of PD-1 and PD-L1 (sPD-L1), blood-based TMB, and immune cell subpopulations [159]. In fact, some of these markers have been shown to be useful in predicting response to ICI treatment [160]. Specifically, low levels of sPD-L1 may correlate with longer survival in advanced NSCLC, multiple myeloma, and renal cell carcinoma [161]. Similarly, complete reduction in ctDNA levels after initiation of therapy was associated with clinical response, while increments or no changes in ctDNA levels were observed in nonresponder NSCLC patients [162]. Blood exosomes and other extracellular vesicles (EVs) are also emerging in the field of oncology for their possible clinical applications. Thus, PD-L1+ exosomes released from tumors can inhibit T-cell activation in the tumor-draining lymph node [96]. In addition, PD-L1 levels in exosomes have been associated with NSCLC progression, but these results were not observed with sPD-L1 in the same study [163]. In turn, EVs are ubiquitous mediators of intercellular communication that transport proteins, metabolites, RNA species, and nucleic acids. They can be found in biological fluids other than blood and can be emitted by tumors with organ-specific immunosuppressive loads to prepare metastatic niches [164]. However, the role of EVs is still poorly studied and requires further investigation and understanding. On the other hand, analysis of multiple antigens at the single-cell level in peripheral blood mononuclear cells (PBMCs) by mass cytometry has recently made it possible to identify certain cellular subpopulations that correlate with a better response to treatment with anti-PD-L1 in melanoma. Studies in melanoma have also described that the frequency of specific cell populations in peripheral blood could also be used to predict the response to anti-PD-1 [165]. In patients with NSCLC specifically, the levels of specific immune cells have also been proposed as response predictors to PD-1/PD-L1 blockade. Thus, CD8+ T-cell density has been reported as a significant predictive and stage-related prognostic factor, while stromal CD8+ TIL density has independent prognostic value. On the other hand, immunosuppressive immune cells such as Tregs, MDSCs, tumor-associated macrophages (TAMs) and neutrophils in the TME have emerged as relevant predictors in this disease. However, the heterogeneity of these populations (i.e., due to differences in CD25 or Foxp3 expression in the case of Tregs) causes them to show discrepancies, making their interpretation difficult, which will require further investigation to reach consistent conclusions in lung cancer. Along these same lines, a high B-cell count and DC aggregates in the tumor are correlated with good clinical outcomes, suggesting that they could have prognostic value in NSCLC [147]. In the case of the immune contexture of SCLC, it is known that it can differ depending on the tumor stage and performance status. Therefore, further research is necessary to understand how variations in immune subpopulations could be key to the ICI response.

In turn, it is well known that the immune system has coevolved with the microbiome that populates the human body, and its composition provides many beneficial functions to its host, including nutrient synthesis, protection against pathogen invasion and regulation of immunological responses to autoantigens [166,167,168]. This suggests that the microbiota plays a critical role in balancing the host immune system between activation and tolerance, since it retains the potential to interfere with innate and adaptive immune responses through different mechanisms. In other words, breaking this relationship could contribute to the reduced effectiveness of immunotherapy in cancer. There is already scientific evidence showing the importance of the gut microbiota in the response to CT and IT and how its alteration due to the concomitant use of antibiotics inhibits the benefit of immune checkpoint inhibitors in advanced cancer, decreasing OS and PFS in NSCLC [140]. Thus, different microbiome clusters have been proposed as predictive biomarkers; however, the strong variability of microbiota according to geographic, dietary and lifestyle differences makes its interpretation difficult [152,169]. In addition, it is important to note that alterations of the microbiome could be analyzed in different biological samples, such as saliva, sputum, bronchial fluids and lung tissue, instead of feces. However, much remains to be done in this regard, not only before starting treatment but also in longitudinal studies to correlate it with the evolution of the response in more or less invasive samples from patients with lung cancer.

## 7. Conclusions

Immunotherapy has had a deep impact on the treatment paradigm for patients with lung cancer, especially with antibody-based immunotherapy. The current state of immunotherapy for lung cancer is mainly aimed at patients with advanced NSCLC. However, some studies are being carried out in patients with SCLC. Despite the promising results, only a lower-than-expected percentage of patients with lung cancer achieve lasting antitumor efficacy with these therapeutic strategies. This encourages the identification of both primary and acquired resistance mechanisms associated with immunotherapy in the hope of assisting in more focused decision-making in clinical practice, as well as the development of new and more effective immunotherapy strategies. An increasing number of biomarkers with potential predictive or prognostic value are being identified, since there is a real need to anticipate the response of patients to immunotherapeutic treatments, specifically anti-PD-1/PD-L1, and to prevent the possible resistance mechanisms involved. Moreover, the clinical benefit of immunotherapy can be adapted not only according to the characteristics of the tumor itself but also the features of each patient. This will allow us to make decisions even with minimally invasive samples for patients. In addition, we predict that the developed and optimized single-cell technology will play a key role in dissecting intratumoral and host heterogeneity and will provide answers to the immunotherapy response.

In summary, the use of cutting-edge technologies, the types of biological samples analyzed, and the development of clinical trials with novel immunotherapeutic strategies will continue to be essential to improve the clinical outcomes of patients with lung cancer. This improvement will be based mainly on the resistance mechanisms identified in current immunotherapies.

## Figures and Tables

**Figure 1 cancers-12-03729-f001:**
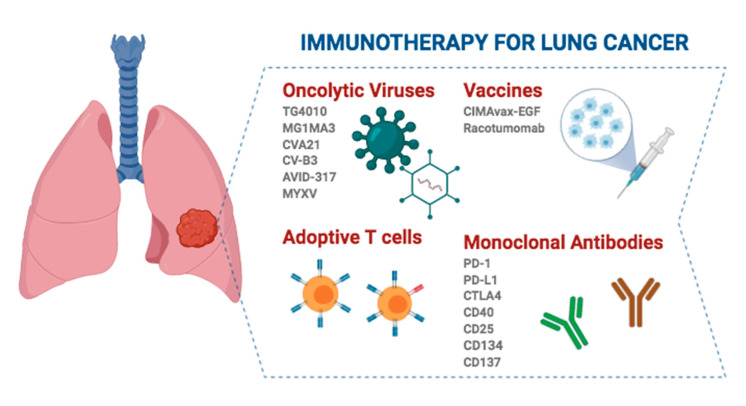
Different strategies and immunotherapeutic agents with clinical application in lung cancer. Created with Biorander.com.

**Figure 2 cancers-12-03729-f002:**
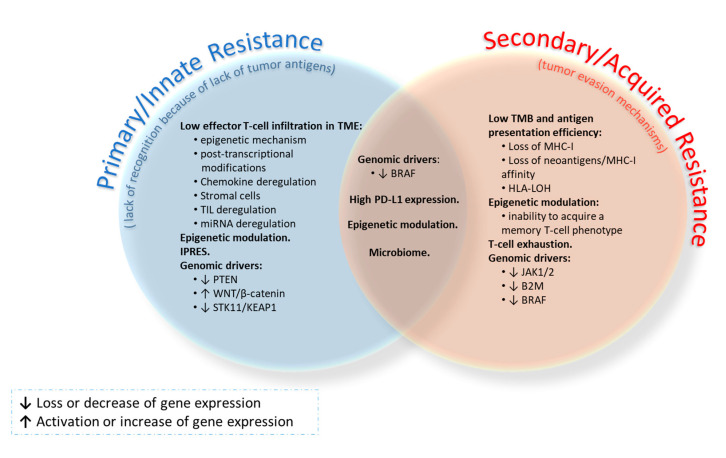
Classification of the most relevant resistance mechanisms to ICI therapies operating in lung cancer.

**Table 1 cancers-12-03729-t001:** NSCLC tumor-associated antigens targeted by monoclonal antibodies that are currently being tested in clinical trials.

Antigen Target	Monoclonal Antibody Name	Clinical Trial ID (https://clinicaltrials.gov)
B7-H3	Enoblituzumab (MGA271)	NCT02475213
BTLA	TAB004	NCT04137900
CD137	BMS-663513	NCT00461110
CD40	APX005M	NCT02482168
	SEA-CD40	NCT02376699
CD44 v6	Bivatuzumab	NCT02204059
CD73	CPI-006	NCT03454451
	Oleclumab	NCT04262388
CEA	Yttrium Y 90 anti-CEA monoclonal antibody cT84.66	NCT00738452
	Yttrium Y 90 anti-CEA monoclonal antibody MN-14 (90Y-hMN-14), indium In 111 anti-CEA monoclonal antibody MN-14	NCT00006458
CEACAM1	CM-24	NCT02346955
c-MET	Sym015	NCT02648724
CSF1R	Cabiralizumab	NCT03502330
CTLA-4	Ipilimumab *	NCT03001882, NCT02350764
	ONC-392	NCT04140526
	REGN4659	NCT03580694
	Tremelimumab	NCT02542293, NCT02000947
DLL3	Rovalpituzumab tesirine	NCT03000257
DLL4	Demcizumab	NCT01189968
EpCAM	Tucotuzumab celmoleukin	NCT00016237
ErbB1/EGFR	Cetuximab	NCT00986674
	Futuximab/modotuximab (Sym004)	NCT02924233
	Necitumumab *	NCT02496663
	Nimotuzumab	NCT01393080
	Matuzumab	NCT00111839
	Panitumumab (ABX-EGF)	NCT00034346
	Pertuzumab	NCT03845270
	SCT200	NCT03808701
ErbB2/HER2	Trastuzumab (Herceptin)	NCT04285671, NCT03505710, NCT03845270
ErbB3/HER3	Seribantumab (MM-121)	NCT02387216, NCT00994123
GDF15	NGM120	NCT04068896
GM2 Ganglioside	BIW-8962	NCT01898156
HGF	Ficlatuzumab (AV-299)	NCT01039948
ICOS	GSK3359609	NCT03693612
	KY1044	NCT03829501
	Vopratelimab	NCT03989362
IGF-1, IGF-2	Xentuzumab	NCT02191891
IGF-1R	Cixutumumab	NCT00778167, NCT00986674
	Dalotuzumab	NCT00951444
	Figitumumab (CP-751,871)	NCT00560573
	Ganitumab (AMG 479)	NCT00807612
IL1RAP	Nidanilimab (CAN 04)	NCT03267316
LAG-3	TSR-033	NCT02817633
LIF	MSC-1	NCT03490669
Mesothelin	Amatuximab	NCT00325494
	Anetumab Ravtansine	NCT03455556
	LMB-100	NCT04027946
CD134	INCAGN01949	NCT02923349
PD-1	BCD-100	NCT03288870
	Budigalimab (ABBV-181)	NCT03000257
	Camrelizumab (SHR-1210)	NCT03527251
	Cemiplimab	NCT03580694
	Dostarlimab	NCT02715284
	Nivolumab *	NCT04043195, NCT04023617
	Pembrolizumab *	NCT04393883, NCT03053856
	Retifanlimab (MGA012)	NCT02475213
	Sasanlimab (PF-06801591)	NCT04181788
	SCT-I10A	NCT04171284
	Serplulimab (HLX10)	NCT04033354
	Sintilimab	NCT03812549
	Spartalizumab	NCT04323436, NCT04000529
	Tislelizumab	NCT03358875
	Toripalimab	NCT04158440, NCT04304248
	Zimberelimab (AB122)	NCT04262856, NCT03629756
PDGF-R α	Olaratumab	NCT00918203
PD-L1	Adebrelimab (SHR-1316)	NCT04316364
	Atezolizumab *	NCT03977467, NCT03645330
	Avelumab	NCT03158883
	Cosibelimab	NCT03212404
	Durvalumab *	NCT02000947, NCT03694236
	Sugemalimab (CS1001)	NCT03789604
	TQB2450	NCT03910127
Phosphatidylserine (Ptd-L-Ser or PS)	Bavituximab	NCT01160601, NCT01138163
PSMA	177Lu-J591	NCT00967577
RAAG12	RAV12	NCT00101972
sCLU	AB-16B5	NCT04364620
SEMA4D	Pepinemab	NCT03268057
TF	MORAb-066	NCT01761240
TGFB	Fresolimumab	NCT02581787
TIM-3	Cobolimab (TSR-022)	NCT02817633
	INCAGN02390	NCT00994123
	MBG453	NCT02608268
TRAIL-R1	TRM-1 (HGS-ETR1)	NCT00092924
TRAIL-R2	Conatumumab (AMG 655)	NCT00534027
VEGF	Bevacizumab *	NCT03836066, NCT03779191
	GB222	NCT04175158
	LY01008	NCT03533127
	QL1101	NCT03195569
VEGFR2	Alacizumab pegol (CDP791)	NCT00152477
	Ramucirumab (IMC-1121B) *	NCT01160744
α5β1 integrin	Volociximab	NCT00666692

* Monoclonal antibodies approved by the FDA for NSCLC.

**Table 2 cancers-12-03729-t002:** SCLC tumor-associated antigens targeted by monoclonal antibodies that are currently being tested in clinical trials.

Antigen Target	Monoclonal Antibody Name	Clinical Trial ID (https://clinicaltrials.gov)
BEC-2	Mitumomab	NCT00037713
CD56	Lorvotuzumab mertansine	NCT00346385
CEA	Yttrium Y 90 anti-CEA monoclonal antibody MN-14 (90Y-hMN-14), indium In 111 anti-CEA monoclonal antibody MN-14	NCT00006347
CTLA-4	Ipilimumab	NCT03575793
	Tremelimumab	NCT02701400
DLL3	89Zr-DFO-SC16.56	NCT04199741
	Rovalpituzumab Tesirine	NCT03000257
EpCAM	Tucotuzumab celmoleukin	NCT00016237
ErbB1/EGFR	Cetuximab	NCT00104910
ErbB2/HER2	Trastuzumab (Herceptin)	NCT00028535
GD2 ganglioside	Dinutuximab	NCT03098030
	MOAB 3F8	NCT00003022
GD3 ganglioside	Mitumomab	NCT00006352
GM2 Ganglioside	BIW-8962	NCT01898156
IGF-1R	Cixutumumab	NCT00887159
	Dalotuzumab	NCT00869752
Lewis-Y	Hu3S193	NCT00084799
PD-1	Budigalimab (ABBV-181)	NCT03000257
	Camrelizumab (SHR1210)	NCT03755115, NCT03417895
	Nivolumab *	NCT03382561
	Pembrolizumab	NCT03319940
	Serplulimab (HLX10)	NCT04063163
PD-L1	Atezolizumab *	NCT03262454
	Durvalumab *	NCT02701400
	TQB2450	NCT04234607
	ZKAB001	NCT04346914
TAA	Bevacizumab	NCT00079040
TIM-3	INCAGN02390	NCT03652077
VEGF	SC-002	NCT02500914

* Monoclonal antibodies approved by the FDA for SCLC.

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
