# Peer review of "Primary and Acquired Resistance to Immunotherapy in Lung Cancer: Unveiling the Mechanisms Underlying of Immune Checkpoint Blockade Therapy"

_cancers, 2020, doi:10.3390/cancers12123729_

Round 1

Reviewer 1 Report

Minor points:

Line 16: Correct Immunooncology to Immuno-oncology

Line 34: Correct TME to tumor microenvironment (TME)

Line 96: Delet “-“ before Despite

Line 165: OX40 is CD134.

Line 173: Correct tumorenvironment (TME) to TME

Line 212: Correct measles to measles virus (MV)

Line 216: Correct GM-CSF to Granulocyte Macrophage colony-stimulating factor

Line 221: add “virus “ between vaccinia and Ankara

Line 241: Correct measles virus (MV) to MV

Line 249: Is “ACT” needed?

Line 253: CAR is not listed in Abbreviations on Line 947.

Line 265: What does “SUPRA” mean ?

Line 298: Add “-“ between CIMAvax and Epidermal

Line 298: Add (EGF) after Factor

Line 306: Add (mAbs) after antibodies

Line 307: Delete (mAbs)

Line 310: Is NK needed?  

Line 318: Is (HAMA) needed?

Line 327: Is (bsAbs) neded?

Line 339 & Line 530: Correct MSI-H

Line 339 & Line 851: Correct dMMR

Line 347: Correct OX40 to CD134

Line 436: Correct IMPOWER-150 to IMpower150

Line 438: Correct to FDA

Line 441: Correct chemotherapy to CT

Line 443: Correct chemotherapy to CT

Line 459: Correct Chek to Check

Line 461 & 462: What’s is ALL

Line 555: Correct “cytotoxic T lumpocytes (CTLs) to CTLs

Line 514: Hard to understand meanings of arrows

Line 521 & Abbreviations: Is ORR needed?

Line 538: Correct Hellman to Hellmann

Line 590: Is CAF needed?

Line 592: What does FASL menan?

Line 636: Correct TH1 to Th1

Line 652: Make sure miRs are correct.

Line 811: Correct tumor microenvironment to TME

Line 846: Delete a NCT02959437.

Line 850: Delete microsatellite instability

Line 873: Correct NSLCC to NSCLC

Line 894: Correct NSLCLC to NSCLC

Some abbreviations on article are not listed in Abbreviations on Line 947.

Check and correct the styles of all references.

Author Response

We thank the reviewer for these valuable suggestions. All typing errors and suggestions have been carefully checked and modified.

Reviewer 2 Report

1: This review mainly describes immune checkpoint suppression including PD-1/PD-L1 for tumor immunotherapy. It focuses on the impact of immune checkpoint inhibitors on the survival of lung cancer patients, and the resistance of immune checkpoints in immunotherapy. But tumor immunotherapy does not only refer to monoclonal antibodies. Therefore, the title should be more accurate, or the author should add more information about other immunotherapies in subsec. 5.

 2: In subsec.4, in addition to chemotherapy, there are many other effective combination immunotherapy approaches, such as radiation or gene therapy. They should be included.

3: In subsec.4, although NSCLC is the main subtype of lung cancer, impact of immunotherapy on the survival of SCLC patients should be enriched.

4: In subsec.5.6. T-cell exhaustion is a very important drug resistance mechanism in lung cancer’s ICI therapy. More references related to T-cell exhaustion should be added.

Author Response

Reviewer 2 comments:

1: This review mainly describes immune checkpoint suppression including PD-1/PD-L1 for tumor immunotherapy. It focuses on the impact of immune checkpoint inhibitors on the survival of lung cancer patients, and the resistance of immune checkpoints in immunotherapy. But tumor immunotherapy does not only refer to monoclonal antibodies. Therefore, the title should be more accurate, or the author should add more information about other immunotherapies in subsec. 5.

We agree with the reviewer on this point, and we have changes the manuscript title to “Primary and Acquired Resistance to Immunotherapy in Lung Cancer: Unveiling the Mechanisms Underlying of Immune Checkpoint Blockade Therapy”

 2: In subsec.4, in addition to chemotherapy, there are many other effective combination immunotherapy approaches, such as radiation or gene therapy. They should be included.

Following the reviewer´s suggestion, we have included:

Line 474: “In addition, much of the current research on NSCLC is focusing on studying the combination of different ITs with each other, as well as CT, RT, and targeted gene and cell-based therapies, with encouraging results”

On the other hand, information on the combination of radiation and immunotherapy has been included in the line 478:

“Several preclinical and clinical studies are also testing the combination of ICI with RT. This combination has been found to have synergistic effects in NSCLC, improving the survival of lung cancer patients without significantly increasing adverse reactions. For example, in the phase I KEYNOTE-001 trial, patients with advanced NSCLC who had received RT prior to pembrolizumab significantly improved their PFS (4.4 vs. 2.1 months) and OS (10.7 vs. 5.3 months), while toxicity remained similar to patients treated with pembrolizumab alone. However, the underlying mechanisms of this combined therapies, possible biomarkers, and optimal therapy parameters - especially the design of RT- have not yet been clarified [78,79].”

In addition, we have added on line 487 information about gene therapy combined with immunotherapy:

“Finally, there are other strategies that hold great promise against cancer, such as CAR T-cell and suicide gene therapies; in 2013, CAR T-cell therapy achieved a response rate of 89% in acute lymphoblastic leukemia and complete responses in acute B lymphoblastic leukemia, so its effect on solid tumors is currently being investigated. In the case of suicide gene therapy, it is based on the use of genes encoding toxic proteins or enzymes capable of transforming a prodrug into a toxin. Typically, adenovirus and herpes virus, among others, are carriers. The use these therapy systems would lead to the sudden and massive presentation of TAA that can be synergistically enhanced by its combination with ICI. The combination of both therapies has demonstrated its antitumor effect in murine models. However, this hypothesis is still in the preclinical stages of development [80].”

The following references have been included according the new text:

  1. Yang, H.; Jin, T.; Li, M.; Xue, J.; Lu, B. Synergistic effect of immunotherapy and radiotherapy in non-small cell lung cancer: current clinical trials and prospective challenges. Precis. Clin. Med. 2019, 2, 57–70, doi:10.1093/pcmedi/pbz004.
  2. Zhou, J.; Huang, Q.; Huang, Z.; Li, J. Combining immunotherapy and radiotherapy in lung cancer: A promising future? J. Thorac. Dis. 2020, 12, 4498–4503; doi: 10.21037/JTD-2019-ITM-001.
  3. Rangel-Sosa, M.M.; Aguilar-Córdova, E.; Rojas-Martínez, A. Immunotherapy and gene therapy as novel treatments for cancer. Colomb Med (Cali). 2017, 30; 48(3):138-147; doi: 10.25100/cm.v48i3.2997.

3: In subsec.4, although NSCLC is the main subtype of lung cancer, impact of immunotherapy on the survival of SCLC patients should be enriched.

We thank the reviewer for pointing this out, and accordingly we have included relevant information regarding the use of immunotherapy on the survival of SCLC in this manuscript.

Line 400: Thus, the 5-year OS is only ~5% for patients with metastatic NSCLC, and between 20-25% or 2% for SCLC, depending on the extent of the disease“.

Line 449: “With regard to SCLC, nivolumab was also tested for pretreated patients with SCLC in the CheckMate-032 trial, obtaining approval from the FDA. This trial showed an objective response rate of 10% with nivolumab and 23% with nivolumab + ipilimumab, with grade 3-4 adverse effects of 14% and 33%, respectively [74]. Soon after, the addition of atezolizumab (anti-PD-L1) to CT in the first-line SCLC treatment in the IMpower133 trial achieved the first OS improvement in decades [70]. This analysis showed that the risk of death decreased by 30% with the combination versus CT alone, without deterioration of the safety profile (median OS: 12.3 vs. 10.3 months; HR=0.70, 95% CI 0.54 to 0.91, p=0.0069). A similar reduction in the death risk was also identified in the phase II CASPIAN trial, which included the PD-L1 inhibitor durvalumab in combination with CT [71]. Thanks to the results of the KEYNOTE-028 and KEYNOTE-158 trials, the FDA approved in 2019 the use of pembrolizumab as monotherapy for patients with metastatic SCLC with disease progression on or after platinum-based CT and at least one other prior line of therapy [74].

Besides, there are several trials currently ongoing evaluating different uses of ICIs is SCLC. Among them, the phase III CheckMate-331 trial (nivolumab vs topotecan / amrubicin), the phase III CheckMate-451 trial (nivolumab vs nivolumab + ipilimumab vs placebo), the phase III KEYNOTE-604 trial (pembrolizumab + platinum / etoposide vs platinum / etoposide), the phase III CASPIAN trial (durvalumab ± tremelimumab + CT vs CT) and the phase III MERU trial (Rova-T + dexamethasone vs placebo (after CT)) [74].”

The following reference has been included according the new text:

  1. Tsiouprou, I.; Zaharias, A.; Spyratos, D. The Role of Immunotherapy in Extensive Stage Small-Cell Lung Cancer: A Review of the Literature. Can Respir J. 2019, 3; 2019:6860432; doi: 10.1155/2019/6860432.

4: In subsec.5.6. T-cell exhaustion is a very important drug resistance mechanism in lung cancer’s ICI therapy. More references related to T-cell exhaustion should be added.

According to the reviewer´s suggestion, we have provided details regarding the T-cell exhaustion, which we hope provide enough insight.

Line 735: “T-cells play a key role in antitumor function and have demonstrated to be crucial for cancer immunotherapy. However, they are not completely effective because part of them go into a dysfunctional state of exhaustion. This promotes uncontrolled growth of tumors [125]. Under physiological conditions, T-cells specifically recognize and react to tumor antigens through their TCRs. Following T-cell priming and tumor localization, the balance between co-stimulation and co-inhibition determines degree of T-cell activation and subsequent immune response [126]. On the one hand, co-stimulation amplifies T-cells activation and enhances CTL proliferation, survival and effector function. On the other, inhibitory receptors, suppressive soluble mediators, cell subsets and metabolic factors from immunosuppressive TME lead to T cell “exhaustion” [125]. The intensity of these different signals depends on parameters such as specific mutations in cancer cells, spatial gradients in tumor composition, and therapy-induced alterations in TME. Collectively, these immunosuppressive signals in TME lead to the intratumoral T-cell exhaustion by influencing the expression of inhibitory receptors, changing metabolic pathways, modifying the epigenetic state, and altering their transcription factor profiles [127]”.

In addition, we have included in line 750 changes in cytokine production as a result of CD8 T-cell exhaustion:

“Functional exhaustion of CD8+ T-cells has been well described in chronic viral infections and in cancer [87]. Exhausted T-cells show poor effector function, express inhibitory receptors, and have altered transcriptional states. They are characterized by a hierarchical loss of proliferation and cytolytic activity, followed by defects in cytokine production and eventual depletion. In a first stage, interleukin-2 (IL-2) production and ex vivo killing capacity are lost, followed by loss of tumor necrosis factor-α (TNF-α) production, and ends with loss of interferon-γ (IFN-γ) and granzyme B (GzmB) production [128]”.

And finally:

Line 774: “Therefore, the scientific community has focused on the development of therapies that enhance T-cell anti-tumor immunity, including adoptive transfer of TILs, endogenous peripheral blood-derived T-cells (ETC), CAR-T, and TCR -engineered T-cells (TCR-T), neoantigen vaccines and checkpoint blockade therapies. To improve the efficacy of these therapies, the dysfunctional state of the T-cells must be reversed. This has led to the investigation of transcriptional regulators as well as metabolic and epigenetic factors as possible targets [125]”.

The following references have been included according the new text:

  1. Zhang, Z.; Liu, S.; Zhang, B.; Qiao, L.; Zhang, Y.; Zhang, Y. T Cell Dysfunction and Exhaustion in Cancer. Front. Cell Dev. Biol. 2020, 11; 8:17; doi: 10.3389/fcell.2020.00017.
  2. Fares, C.M.; Van Allen, E.M.; Drake, C.G.; Allison, J.P.; Hu-Lieskovan, S. Mechanisms of Resistance to Immune Checkpoint Blockade: Why Does Checkpoint Inhibitor Immunotherapy Not Work for All Patients? Am. Soc. Clin. Oncol. Educ. B. 2019, 147–164, doi:10.1200/edbk_240837.
  3. Thommen, D.S.; Schumacher, T.N. T Cell Dysfunction in Cancer. Cancer Cell 2018, 9; 33(4), 547-562; doi: 10.1016/j.ccell.2018.03.012.
  4. Jiang, Y.; Li, Y.; Zhu, B. T-cell exhaustion in the tumor microenvironment. Cell Death Dis. 2015, 18; 6(6):e1792; doi: 10.1038/cddis.2015.162.

Reviewer 3 Report

I highly appreciate the author's effort in compiling all the relevant information in the review titled: "Primary and Acquired Resistance to Immunotherapy  in Lung Cancer."

Authors have put a tremendous effort into organizing and compiling all the information related to lung cancer resistance to immunotherapy. I specifically liked the table compiling all the clinical trials related to lung cancer treatments. Overall, this article's information is very crucial for other researchers in the field.  I will recommend this scientifically sound and well-written review to be published as it is without any further revisions.

Author Response

We appreciate the recognition of our work by the reviewer.